# AFM Analysis of a Three-Point Flexure Tested, 3D Printing Definitive Restoration Material for Dentistry

**DOI:** 10.3390/jfb14030152

**Published:** 2023-03-10

**Authors:** Maximilian N. Sandmair, Christoph Kleber, Dragan A. Ströbele, Constantin von See

**Affiliations:** 1Research Center for Digital Technologies in Dentistry and CAD/CAM, Department of Dentistry, Faculty of Medicine and Dentistry, Danube Private University, 3500 Krems, Austria; 2Department of Medicine, Faculty of Medicine and Dentistry, Danube Private University, 3500 Krems, Austria

**Keywords:** 3D printing, DLP, CAD/CAM, AFM, surface analysis, surface roughness, biomaterials

## Abstract

Background: Three-dimensional printing is a rapidly developing technology across all industries. In medicine recent developments include 3D bioprinting, personalized medication and custom prosthetics and implants. To ensure safety and long-term usability in a clinical setting, it is essential to understand material specific properties. This study aims to analyze possible surface changes of a commercially available and approved DLP 3D printed definitive restoration material for dentistry after three-point flexure testing. Furthermore, this study explores whether Atomic Force Microscopy (AFM) is a feasible method for examination of 3D printed dental materials in general. This is a pilot study, as there are currently no studies that analyze 3D printed dental materials using an AFM. Methods: The present study consisted of a pretest followed by the main test. The resulting break force of the preliminary test was used to determine the force used in the main test. The main test consisted of atomic force microscopy (AFM) surface analysis of the test specimen followed by a three-point flexure procedure. After bending, the same specimen was analyzed with the AFM again, to observe possible surface changes. Results: The mean root mean square (RMS) roughness of the segments with the most stress was 20.27 nm (±5.16) before bending, while it was 26.48 nm (±6.67) afterward. The corresponding mean roughness (Ra) values were 16.05 nm (±4.25) and 21.19 nm (±5.71) Conclusions: Under three-point flexure testing, the surface roughness increased significantly. The *p*-value for RMS roughness was *p* = 0.003, while it was *p* = 0.006 for Ra. Furthermore, this study showed that AFM surface analysis is a suitable procedure to investigate surface changes in 3D printed dental materials.

## 1. Introduction

The history of 3D printing dates back to the early 1980s, when Chuck Hull, a researcher at 3D Systems, invented stereolithography (SLA) [1,2,3]. SLA was the first 3D printing process that involved curing photopolymer resin using ultraviolet light, building up a 3D object layer by layer [2,3]. In 1986, Hull filed a patent for the SLA process and 3D Systems started commercializing the technology [2,3]. At the start of the 21st century, 3D printing was still a niche technology that was primarily used for prototyping and creating models by product designers and engineers [2,3]. However, advancements in technology and a decrease in cost eventually led to the growth of the consumer and small business 3D printing market [2,3]. By the 2010s, 3D printing had become more accessible and was being utilized in a variety of industries, including aerospace, fashion, and healthcare [2,3]. In addition to SLA and photopolymerization, various other 3D printing technologies have emerged in recent decades [2,3]. These include Fused Deposition Modeling (FDM), which involves melting and depositing plastic filament on a build platform [4]. Powder bed fusion, which uses a laser to sinter or melt powdered material [5]. Material jetting, which is a process where melted material is jetted onto a build platform [5]. Sheet lamination, that uses thin sheets of material cut by lasers or a sharp blade and glued together [6]. Lastly, 3D bioprinting, a method that creates functional human tissue for medical applications, such as drug testing and transplants [3,7,8]. Three-dimensional printing has been utilized in various aspects of medicine beyond bioprinting, including custom prosthetics and implants, surgical planning and training, and personalized pharmaceuticals [9]. With 3D printing, personalized medication, such as pills and patches, tailored to an individual’s specific needs can be created [3,7]. Three-dimensional printing also creates anatomically accurate models of patient-specific anatomy, useful for complex surgical planning and medical professional training [3,7,10,11]. The technology is also being applied to prosthetics and implants, that can be custom-fit to a person’s needs, making them more comfortable and functional [7,12]. Three-dimensional printing in dentistry is a rapidly growing field that is revolutionizing the way dental professionals approach patient care [10,11,13,14,15,16,17,18,19]. It involves the use of computer-aided design (CAD) software to create 3D models of dental structures and the use of 3D printers to produce physical models, surgical guides, and even dental prosthetics such as crowns, bridges, and implants [11,13,15,16,17,18,19,20,21]. The ability to produce precise, customized dental prosthetics in a fraction of the time it takes using traditional fabrication methods has numerous benefits for both the dentist and the patient [11,13,15,16,17,18,19]. For dentists, 3D printing allows for more accurate assessments of patient anatomy and the creation of a virtual treatment plan [11,13,15,16,17,18,19]. For patients, 3D printing enables chair-side fabrication, which reduces waiting time, the number of visits required and eliminates the need for temporary restorations, increasing comfort and satisfaction [11,13,15,16,17,18,19]. Another advantage of 3D printing in dentistry is the use of digital impressions, which can be used in place of traditional, physical impressions [11,13,15,16,17,18,19]. Digital impressions are captured using a specialized intraoral scanner, which captures high-resolution images of the tooth structure and surrounding tissues [11,13,15,16,17,18,19]. These images are then processed into a 3D model, which can be used to create a physical model of the patient’s dentition [11,13,15,16,17,18,19]. This process not only eliminates the need for physical impressions but also reduces the margin of error inherent in traditional impression methods [11,13,15,16,17,18,19]. Despite the numerous advantages of 3D printing in dentistry, there are some limitations to consider. One of the main challenges is the initial cost, as 3D printing technology and materials can be expensive [11,13,15,16,17,18,19]. Furthermore, the training and experience required to use 3D printing technology effectively, can also be a barrier for some dental professionals [11,13,15,16,17,18,19]. Lastly the materials are required to possess specific mechanical properties to be suited for clinical application [11,13,15,16,17,18,19]. Currently, 3D-printed materials may be applied as long-lasting temporary and permanent restorations in a clinical setting [22,23]. In order to meet the mechanical and chemical requirements, thorough testing is required prior to in-vivo application. Although clinical application cannot be fully simulated in vitro, it is nevertheless possible to explore in vitro material-specific properties that are indispensable for a foundational comprehension of the material [24,25,26]. Since the 3D printed resin for permanent restorations has just been developed recently, there is only a limited range of data and studies available on its properties; none of which use atomic force microscopy (AFM) for analysis [13,14]. This study aims to analyze the surface changes of a commercially available and approved 3D printed definitive material using an AFM before and after three-point flexure testing. Furthermore, it explores whether AFM is a feasible method for examination of these materials in general. The null hypothesis (H0) is that after three-point flexure testing there is no difference in surface roughness.

## 2. Materials and Methods

The present study consisted of a pretest followed by a main test. The pretest was performed to obtain a break force for the specimens, which was subsequently used to determine the force for the main test. The main test consisted of a preliminary AFM surface analysis of the specimen followed by a three-point flexure procedure. After bending, the specimens were analyzed by AFM again to determine possible surface changes.

The test specimens were made of the ceramic filled hybrid material for 3D printing of permanent restorations VarseoSmile Crown plus (BEGO, Bremen, Germany) [22]. Based on ISO 178 and ISO 899-2, the dimensions were chosen to be 80 mm by 10 mm by 4 mm.

Three specimens were prepared; two for a preliminary test and one for the final analysis. For statistical evaluation the area observed with the AFM was divided into nine segments. The samples were 3D modeled in Autodesk Netfabb Premium 2021.1 (Autodesk, San Rafael, CA, USA) (Figure 1). The printing layers were perpendicular to the direction of loading. Post-processing followed the manufacturer’s instructions and consisted of an ultrasonic bath in isopropanol, followed by light curing in the Otoflash (BEGO, Bremen, Germany) with two cycles of 1500 flashes each. Supports were removed with a cutting wheel and dental side cutters. The samples were then sandblasted using the 50 µm glass bead blasting material Perlablast micro (BEGO, Bremen, Germany) at a pressure of 1.5 bar. Afterward, the surface was polished using a pumice stone and polishing compound with care not to overheat the workpiece.

Two of the prepared test samples were loaded until failure in the universal testing machine Z010 (Zwick/Roell, Ulm, Germany) to obtain a basic understanding of the fracture forces of the specimen.

The specimen for the main test was analyzed under the atomic force microscope CoreAFM (Nanosurf AG, Liestal, Switzerland) (Figure 2). In total, four regions with a size of 96.8 µm were chosen. The four regions were on the tension side of the workpiece. The tension side is on the bottom, opposite the loading pin (Figure 3). The first region was directly in the middle of the specimen where loading occurred. The other remaining measurement regions were evenly spaced outwards in 5 mm steps. The AFM image consisted of 512 points per line with a measuring speed of 1.4 s per line. The measurement regions were numbered from one to four, starting from the middle and going outwards. Tapping mode was applied to avoid a possible movement of surface features. After imaging, the test sample underwent the three-point flexure test in the Z010 (Figure 4). The test consisted of a loading phase and a holding phase. The target force for the three-point bending procedure was calculated from the results of the pretest. First, 80% of the break force was used as a starting point for the bending procedure. The resulting force caused breakage of the workpiece in the bending procedure, so the target bending force was adjusted to 75%. This resulted in 37 Newtons achieving a strong bend without failure (Figure 5). The speed to approach the target force was 0.05 mm/min. This position was held for 19 h. After loading, the sample was analyzed in the same regions as before in the AFM. The images were edited in Gwyddion^®^ (open source software solution) for data analysis [27,28]. Each image’s data was leveled by mean plane subtraction. Horizontal scars were corrected, and the minimum data value was shifted to zero. Subsequently, the data was cut to obtain two images identical in size and location from each region. Each region’s root mean square (RMS) roughness and mean roughness (Ra), before and after bending was compared. As the first region, which was closest to the point of loading, showed the highest difference in roughness, it was chosen for further examination. The cut images of the first region were divided into nine segments for statistical evaluation. The RMS roughness, as well as the Ra of each segment before and after bending was calculated and exported. This resulted in nine pairs of before and after images identical in size and location. Whether the data was normally distributed and if there was a significant mean difference between the pairs of measurements was determined with a Shapiro–Wilk test followed by a paired *t*-test using SigmaPlot 13.0 (Systat Software Inc., San Jose, CA, USA).

## 3. Results

Four regions of the specimen were analyzed before and after bending. The largest change in surface roughness occurred in region 1. Therefore, it was chosen for further examination. The cut images of the first region were divided into nine segments for statistical evaluation. The AFM images are depicted in Figure 6, Figure 7, Figure 8 and Figure 9. The nine segments of the first region are visualized in Figure 10.

### 3.1. Surface Roughness of Regions 1–4

The RMS roughness of the first region before loading was 27.60 nm, while it was 41.59 nm after loading. For the second region, it was 48.80 nm and 47.06 nm. For the third, it was 27.88 nm and 33.19 nm. For the fourth, it was 36.06 nm and 35.16 nm.

The Ra of the first region before loading was 21.98 nm, while it was 34.60 nm after loading. For the second region, it was 39.37 nm and 36.96 nm. For the third, it was 21.54 nm and 25.41 nm. For the fourth, it was 29.04 nm and 27.85 nm.

### 3.2. Surface Roughness of Segments 1–9 Obtained from Region 1

The RMS roughness of the first segment before loading was 16.33 nm, while it was 22.65 nm after loading. For the second region, it was 23.04 nm and 29.86 nm. For the third, it was 31.08 nm and 34.58 nm. For the fourth, it was 15.96 nm and 18.42 nm. For the fifth, it was 18.23 nm and 19.97 nm. For the sixth, it was 19.06 nm and 19.74 nm. For the seventh, it was 24.71 nm and 36.31 nm. For the eighth, it was 15.09 nm and 26.50 nm. For the ninth, it was 18.94 nm and 30.31 nm.

The Ra of the first segment before loading was 12.85 nm, while it was 18.06 nm after loading. For the second region, it was 18.69 nm and 24.15 nm. For the third, it was 25.64 nm and 28.52 nm. For the fourth, it was 12.66 nm and 14.08 nm. For the fifth, it was 14.42 nm and 14.76 nm. For the sixth, it was 15.06 nm and 15.64 nm. For the seventh, it was 17.67 nm and 27.71 nm. For the eighth, it was 11.89 nm and 21.87 nm. For the ninth, it was 15.53 nm and 25.96 nm.

### 3.3. Statistical Evaluation

In the statistical evaluation, the confidence level was set to 95%. The Shapiro–Wilk test was passed with a *p*-value of 0.137 for the RMS roughness and a *p*-value of *p* = 0.114 for Ra. The paired *t*-test concluded that the change that occurred with the segments is greater than would be expected by chance; there is a statistically significant change of RMS roughness (*p* = 0.003) as well as Ra (*p* = 0.006). The results are summarized in Figure 11 and Figure 12.

## 4. Discussion

The mean RMS roughness of the segments with the most stress was 20.27 nm (±5.16 nm) with a standard error (SE) of 1.72 nm before bending, while it was 26.48 nm (±6.67 nm) with a SE of 2.22 nm afterward. The corresponding Ra values are 16.05 nm (±4.25 nm) with a SE of 1.42 nm and 21.19 nm (±5.71 nm) and a SE of 1.90 nm. There were no outliers.

The test specimen’s dimensions were chosen based on ISO 178 and ISO 899-2, as those standards are used to determine the flexural properties of plastics. As 3-D printing in dentistry is relatively new, there is no standardized test specifically for 3D printing yet [13,14]. Therefore, the procedure was modified to fit the needs of the present investigation. Nold et al. [29] have shown that the printing direction can significantly impact the properties of 3D printed objects. Whether that is the case for VarseoSmile Crown plus needs further investigation, specifically in context with surface analysis. As the printing layers could turn out to be a potential mechanical weak point of the material, the printing layers were orientated perpendicular to the loading direction. Post-processing followed the manufacturers instruction to obtain the same finish as used in the clinical reality [22].

A set of fracture values was obtained with a preliminary test. This determined a target force for the three-point flexure procedure. The goal was to approximate a value near the breaking point of the specimen without reaching it. This ensured the most prominent effect of bending without risking failure of the material. In general, deformation can be categorized into three phases. The elastic phase, the plastic or irreversible phase and lastly the fracture or failure of the workpiece. Elastic deformation and plastic deformation are two types of deformation that occur in materials when subjected to an external load [30,31,32,33]. Elastic deformation refers to a temporary change in shape of a material under an applied load, which returns to its original shape when the load is removed [30,32,33]. This type of deformation occurs within the elastic limit of the material, where the material’s stress, which is the force per unit area, is proportional to its strain, which is the change in dimension per unit length [30,33]. The proportionality constant between stress and strain is called the modulus of elasticity, which is a measure of a material’s stiffness [30,33]. VarseoSmile CrownPlus, as a polymeric material, is composed of long chains of repeating molecular units that are held together by weak intermolecular forces [22,30,33,34]. When a plastic material is subjected to an external load, its molecular chains are subjected to stress and change their positions slightly [30,32,33]. If the load is removed, the molecular chains return to their original positions and the material returns to its original shape [30,32,33]. The amount of deformation that a plastic material experiences is proportional to the magnitude of the applied load, with larger loads resulting in larger deformations [30,33]. The modulus of elasticity of a plastic material depends on its chemical composition and molecular structure, as well as its processing history [30,32,33]. For example, materials with high molecular weight and strong intermolecular forces tend to have a higher modulus of elasticity and be stiffer [30,33]. The modulus of elasticity is an important factor in determining the suitability of a plastic material for a given application, as materials with a high modulus of elasticity are typically more rigid and resistant to deformation [30,32,33]. Plastic deformation, on the other hand, is defined as the ability of a solid material to undergo a permanent non-reversible dimensional change in response to applied forces [30,32,33]. Plastic deformation can be observed in most materials. However, the physical mechanisms can vary widely. For example, in metals plasticity happens at a crystalline scale in form of slip and twinning, whereas in more brittle materials like rock and bone, plasticity can be caused by slip and microcracks [33,35]. When a polymer is subjected to an applied load, the intermolecular forces can be broken, causing the molecular chains to slip past one another and reorient [30,33]. The extent of plastic deformation in a plastic material depends on several factors, including the material’s yield strength, the type of loading, the temperature and the strain rate [30,32,33]. Yield strength is the amount of stress required to produce a predetermined amount of permanent strain [30,32,33]. It is an important factor in determining a material’s ability to maintain its shape under stress. The type of loading, temperature, and strain rate can all affect the material’s yield strength, with increased temperature and strain rate typically leading to increased plastic deformation [30,33]. Plastic deformation in plastics can also be influenced by cross-linking agents providing bridges between linear macromolecules to form a three-dimensional network [30,33]. Another factor is the presence of defects, such as voids, inclusions, and cracks, which can act as stress concentrators and increase the likelihood of plastic deformation [30,33].

The purpose of this investigation was to present preliminary findings of a physical phenomenon. Contrary to SEM the use of the AFM made it possible to make no structural alterations and therefore limiting factors that could have influenced the outcomes. While the current study used a limited number of specimens, additional research can provide more comprehensive data and offer a deeper understanding of this phenomenon, by increasing the sample size or chemical analysis. Afterward, future analytical simulations like FEM could provide further valuable insight.

The aim of the present investigation was to reach the plastic phase of deformation. This was accomplished as the test specimen was visibly and irreversibly bent. Because irreversible deformation during mastication could result in insufficient restorations, irreversible deformation without failure is not desirable in dentistry. This results in a generally small window of plastic deformation in dental materials [30,32,33]. Therefore, in order to reproducibly reach the phase of plastic deformation, getting close to the breaking point is important. First, 80% of the break force was used as a starting point for the bending procedure. The resulting force caused breakage of the workpiece in the bending procedure, so the target bending force was adjusted to 75%. With further research, a more optimized and standardized procedure could be created.

To check if there is a difference in roughness across the sample, four measurement regions from the point of highest stress with additional subsequent steps away from it were analyzed. As the first region showed the highest difference after bending, further investigations could reduce the distance between the measurement regions to see how the specimen is affected between the first and second region that was analyzed in this study. This region was then chosen for further examination. The cut images of the first region were divided into nine segments for statistical evaluation. AFM is a type of scanning probe microscopy (SPM) technique used to image, analyze and manipulate surfaces at a nanoscale resolution [36,37]. Nowadays, it is a frequently used tool to analyze conventional materials in dentistry [38,39,40,41,42,43]. It works by using a sharp probe tip attached to a flexible cantilever, which is moved in close proximity to the sample surface in a raster scanning motion [37,44]. The probe tip scans the surface and the deflection of the cantilever is measured using a laser and a photodetector [37,44]. The probe tip interacts with the sample surface, either by tapping on the surface or by holding the cantilever in close proximity to the sample. This interaction generates a force between the probe tip and the sample [37,44]. The deflection of the cantilever is proportional to the magnitude of this interaction force [37]. By measuring the cantilever deflection, the topography of the sample can be reconstructed [37]. AFM can be used in a number of imaging modes, including tapping mode, contact mode, and non-contact mode, each of which has its own advantages and disadvantages [36,37,44]. In this study tapping mode was chosen as the imaging mode. Tapping mode is the most frequently used imaging method in Atomic Force Microscopy [36]. This mode operates by holding the probe tip above the sample surface, avoiding direct contact between the tip and the sample. This method provides high-resolution images, with a lateral resolution of a few nanometers [37,44]. In addition, tapping mode is a non-destructive imaging method that reduces the wear of the probe tip and increases the stability of the sample due to the mitigation of damaging lateral forces [36,37]. However, there are also some disadvantages to using tapping mode. One of the main limitations of this mode is the slower imaging speed compared to other modes, such as contact mode [36]. Additionally, the quality of the images obtained with tapping mode may not be as high as those obtained with other modes, especially for samples with rough or highly textured surfaces [36]. Furthermore, tapping mode has a limited imaging range, and it is difficult to image features with vertical dimensions less than a few nanometers [36]. Finally, there is an increased noise in the data obtained with tapping mode, as the laser and photodetector are used to detect small fluctuations in the cantilever deflection [36,37]. In addition to imaging, AFM can also be used for a range of other applications, including measuring mechanical properties of samples, investigating chemical interactions at the nanoscale, and nanoscale manipulation and fabrication [37,44].

The loading phase of the test was conducted with a speed of 0.05 mm/min to not overload the sample too quickly and avoid fracture. The holding phase was chosen to be 19 h. “Creep is defined as the time-dependent plastic strain of a material under a static load or constant stress.” [30] (p.24). The existence of in vivo creep of dental materials, especially during intake of hot food followed by mastication or long span bridges, is a fact generally understood [30]. Therefore, the aforementioned time frame gave the material time to bend, maximizing changes in the material.

The images were edited in Gwyddion. The data was leveled, and error lines were removed to get a more accurate representation of the sample. The images were cut and resized to obtain matching pairs of images identical in size and position. Alongside the commonly used Ra the RMS roughness was also included, as it has a higher sensitivity to larger peaks and valleys. Ra is calculated as the arithmetic average of the measured roughness profiles, while RMS roughness is the root mean square of the measured roughness profiles.

The Ra of the segments with most stress was 16.05 nm (±4.25 nm) before bending and 21.19 nm (±5.71 nm) afterward. Alharbi et al. [45] have analyzed commercially available polymer-based ceramics materials. Their roughness values were lower, ranging from 7.75 nm (±0.1 nm) to 13.16 nm (±0.5 nm) [45]. Alharbi et al. [45] also analyzed a new polymer infiltrated ceramic network (PICN) material, with mean Ra values similar to the ones in the present investigation. Their Ra ranged from 18.94 nm (±1.1 nm) to 31.21 nm (±3.6 nm) depending on the processing method used. However, there are currently no studies that compare AFM analyzed 3D printed dental materials before and after flexure testing.

In dentistry, plastics can be modified with fillers to enhance the elastic modulus, increase tensile strength, hardness and wear resistance [46,47,48,49,50]. These fillers might play a significant role in surface changes in the bending procedure. Further research is needed to investigate the origin of the present increase in surface roughness and its repeatability. Increased surface roughness is undesirable as it can lead to increased bacterial adhesion [51,52,53,54,55,56,57,58,59,60,61,62]. A Sa threshold of 200 nm has been reported as clinically acceptable. Above this level, biofilm formation increases, while below this level, the surface roughness has no significant impact on biofilm formation [52,54,63]. In clinical use, a rougher surface also leads to a matt and less aesthetically pleasing finish [61,62,64,65,66]. Furthermore, increased surface roughness could lead to increased discoloration [64,65,66] and wear of the opposing teeth [67]. In the present analysis before and after bending, every measured segment stayed below the aforementioned threshold of 200 nm. Nevertheless, the mean Ra rose from 16.05 nm (±4.25) to 21.19 nm (±5.71); a 132% increase.

Further studies need to be conducted to learn about the clinical consequences of the measured increase in surface roughness and its clinical effects in combination with the present material. Further studies are also needed to understand whether the material undergoes similar changes in vivo. As the masticatory force, time period of stress, temperature and other factors in vivo differ from this study, the reported surface changes cannot be compared directly to possible changes in vivo. Nevertheless, this investigation shows that flexural stress can cause a significant change in the material’s properties. Furthermore, this study showed that AFM surface analysis is a suitable procedure for investigating surface changes in 3D printed dental materials.

## 5. Conclusions

The AFM analysis of the three-point flexure tested, 3D printing definitive restoration material VarseoSmile Crown plus for dentistry resulted in the following outcomes:Increased surface roughness after plastic deformation;Higher roughness than conventional materials, but still clinically acceptable before and after flexure testing;AFM surface analysis is suitable for investigating surface changes in 3D printed dental materials.

Further research is needed in the following areas:Significance of printing direction and its impact on the properties of 3D printed objects produced from VarseoSmile Crown plus;Creation of an optimized and standardized procedure to reach plastic deformation in 3D printed dental materials;Reduced distance between measurement regions to see how much of the workpiece is affected;Impact of fillers in surface changes in the bending procedure;Clinical consequences of the measured increase in surface roughness and its clinical effects in combination with VarseoSmile Crown plus;Material changes in vivo.

## Figures and Tables

**Figure 1 jfb-14-00152-f001:**
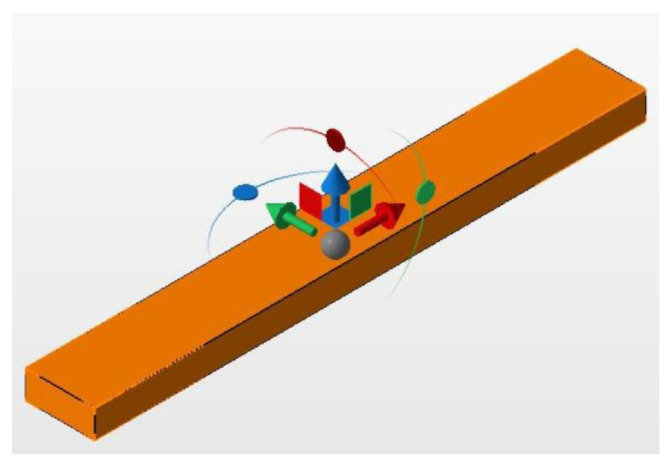
Specimen during design.

**Figure 2 jfb-14-00152-f002:**
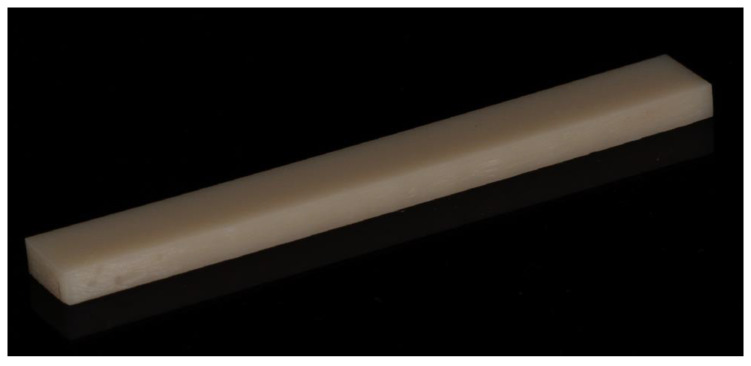
Finished specimen for the main test.

**Figure 3 jfb-14-00152-f003:**
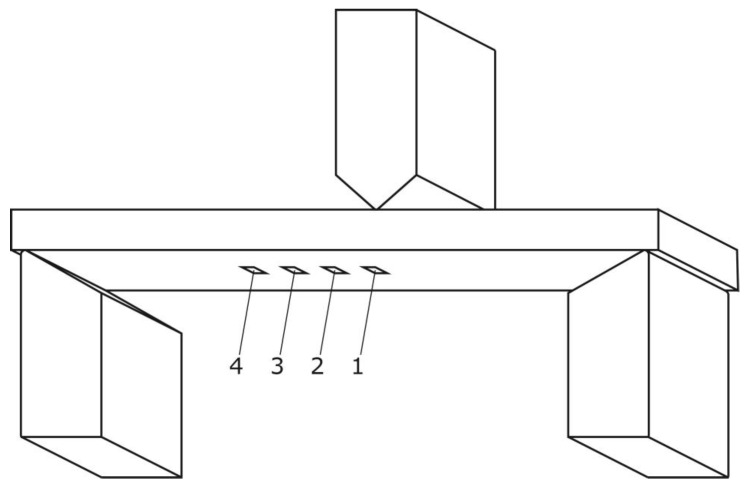
Schematic illustration of the three-point flexure test setup and the regions for AFM analysis.

**Figure 4 jfb-14-00152-f004:**
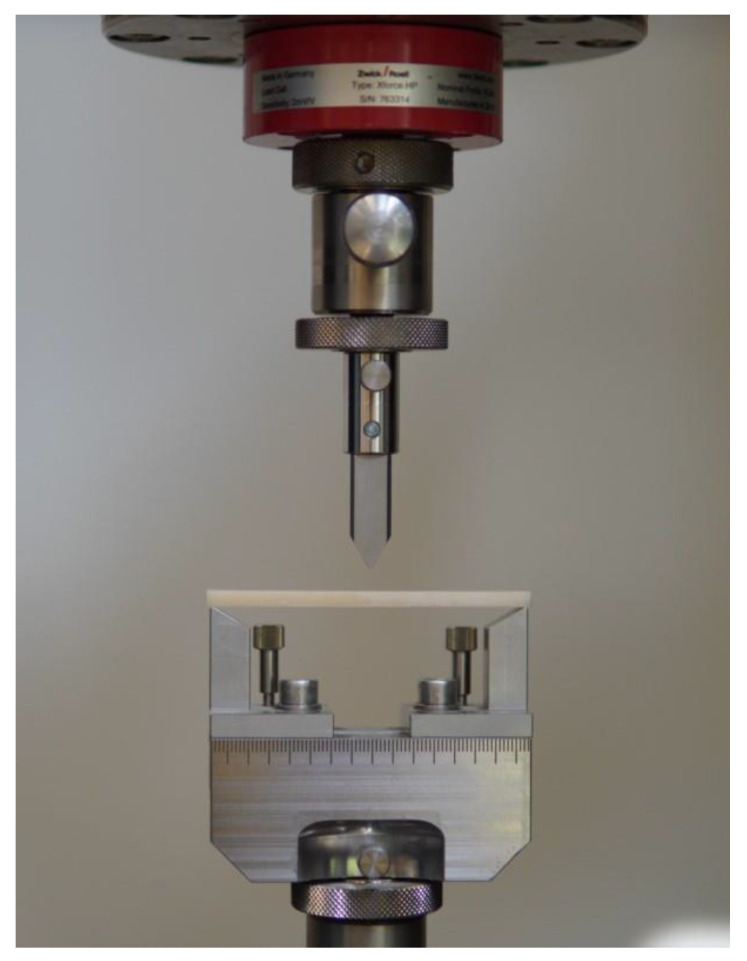
Three-point flexure test setup.

**Figure 5 jfb-14-00152-f005:**
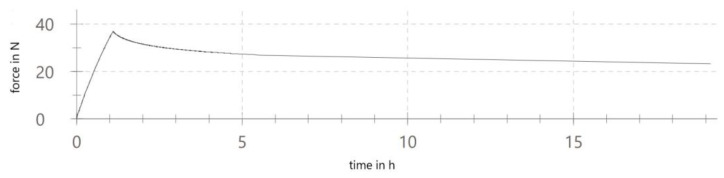
Three-point flexure test graph.

**Figure 6 jfb-14-00152-f006:**
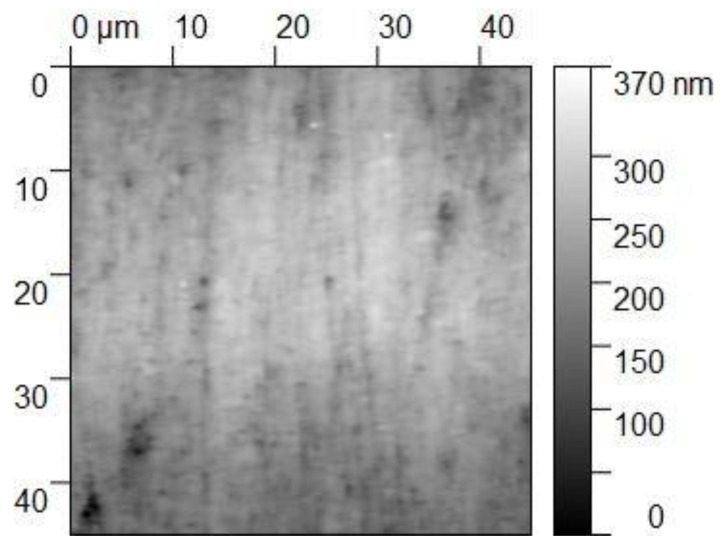
The 45 × 45 µm^2^ TM-AFM image of region 1 before bending. The z-range is 370 nm from black to white.

**Figure 7 jfb-14-00152-f007:**
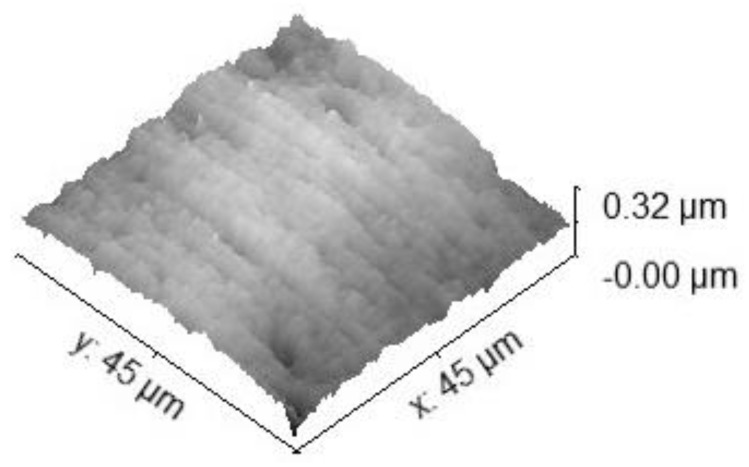
Three-dimensional 45 × 45 µm^2^ TM-AFM image of region 1 before bending. The z-range is 320 nm from black to white.

**Figure 8 jfb-14-00152-f008:**
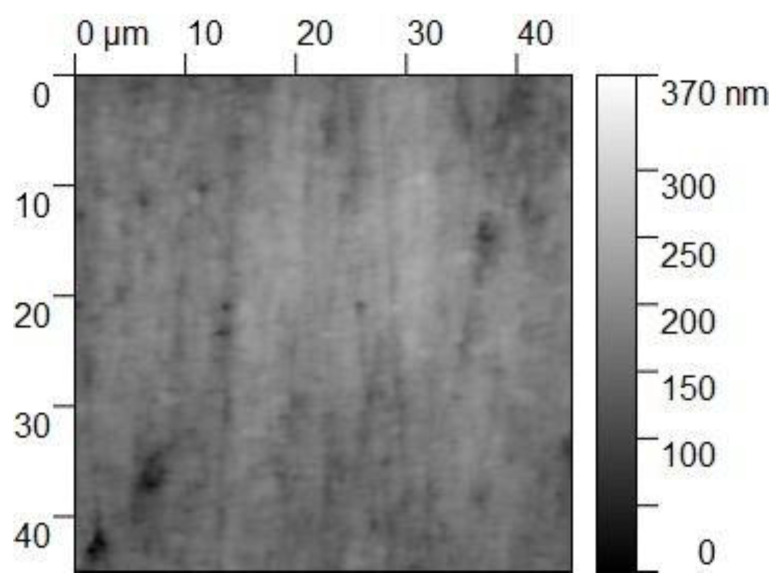
The 45 × 45 µm^2^ TM-AFM image of region 1 after bending. The z-range is 320 nm from black to white.

**Figure 9 jfb-14-00152-f009:**
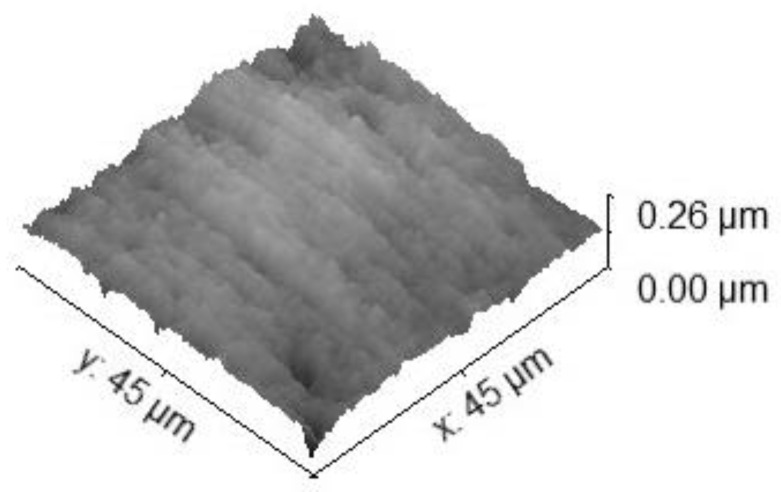
Three-dimensional 45 × 45 µm^2^ TM-AFM image of region 1 after bending. The z-range is 320 nm from black to white.

**Figure 10 jfb-14-00152-f010:**
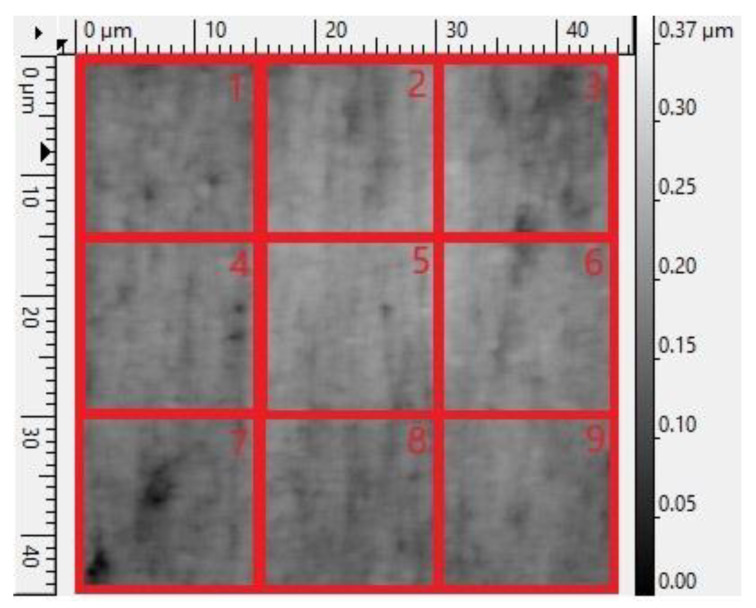
Schematic illustration of the 9 segments for statistical evaluation.

**Figure 11 jfb-14-00152-f011:**
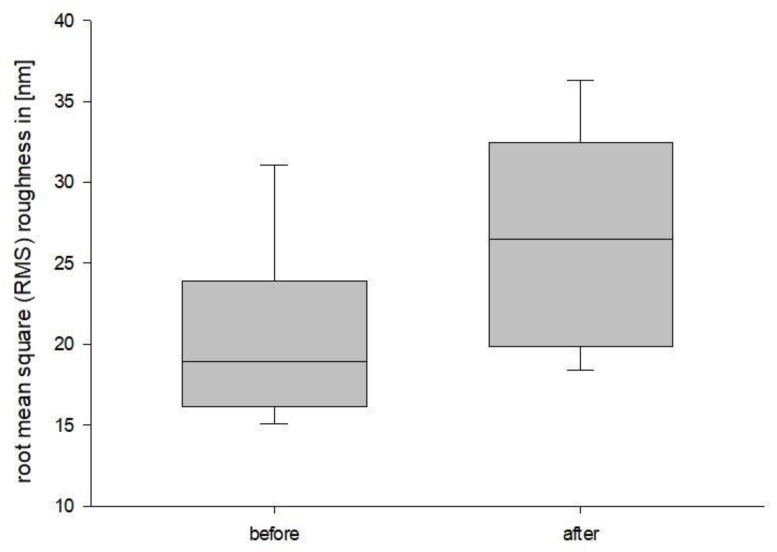
RMS roughness boxplot.

**Figure 12 jfb-14-00152-f012:**
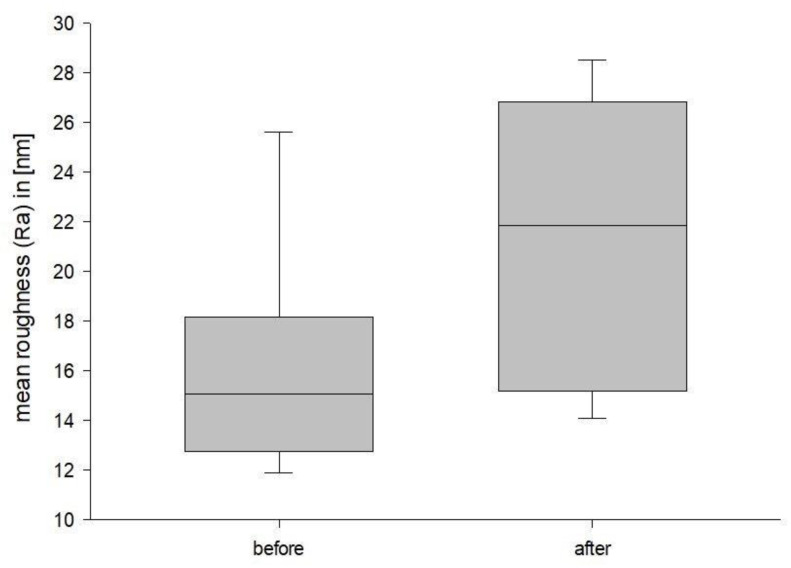
Ra Boxplot.

## Data Availability

Data is contained within the article.

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
