# Peer review of "AFM Analysis of a Three-Point Flexure Tested, 3D Printing Definitive Restoration Material for Dentistry"

_jfb, 2023, doi:10.3390/jfb14030152_

Round 1

Reviewer 1 Report

Dear Authors,

Thank you for submitting this manuscript. I think this paper is quite interesting, because it refers to a very important and well-developed topic, which is 3D printing in dentistry. I would like to suggest some points to the authors:

1. The abstract should include a short statement on the current research gap and question to show why this study is unique and worthy of publication.

2. In the introduction, the references are not in the correct order of appearance; they are repeating in the whole section [2,3,5], please make revisions and add more references.

3. Lines 40 – 46 – This sentence is too long, please revise.

4. Line 48 – Please add more references.

5. Line 57 – 78 – Again you are citing the same reference [5], please add different references.

6. In the introduction, please add a graph/picture on the design and preparation of the samples.

7. The number of test specimens used in the study is only three? – do you think this will be enough for statistical validation, please clarify.

8. Please add a real image of the test sample to the Materials and methods section.

9. The authors should add the null and working hypotheses and highlight them by adding "H0" and "H1"

10. In the Discussion section, please note that the reference goes directly after the name of the Author, when he / she was mentioned, e.g. "Nick et al. [23]" and this should be corrected in the whole text.

11. Lines 213-214 - please combine the short sentences.

12. In the Conclusion section, please describe the significance of this study. In addition, “The P-338 value of the RMS roughness was P=0.003, while it was P = 0.006 for Ra” cannot be regarded as a conclusion, please describe the actual meaning of the changes instead of the change of P value.

13. The authors should summarize the significant findings in bullets for clarity in the Conclusion section.

14. The authors should revise the references; the name of the journal should be in italic and the year of publication in bold (Check instructions for authors).

Thank you in advance for all the corrections. Good luck!

Reviewer 2 Report

The topic is relevant to current context but needs to incorporate all the modifications suggested in the below mentioned form-

             Abstract underscores the content with 194 words (limit of words is up to 300, still there is scope for enhancing with test results). Currently there is no connectivity to the reader why this work is being carried out? Significance of the work? Need of the work has to be explained in the early part of abstract

             As per journal guidelines 3 to 10 keywords need to be used but still there is scope to include any critical keywords

             References should be mentioned as per the journal guidelines

             Total number of references cited 42. Among there is hardly any article cited of 2022. Only 4 articles of 2021, in which again only two articles in introduction section seems to be not fair for a repute of this quality journal but the clarity can be achieved with more addition of articles of recent.

             Authors have to restrict the self-citations. Which looks to be not a good practice

             The plagiarism is 15% and expected to be less than 10% for quality journals (report attached for reference)

            There is still lot of scope for improvising the introduction section as it talks limited in terms of the current trends in 3D printing, AFM, CAD/CAM and Surface analysis. There are plenty of recent articles to support few of them have been listed in below. You are free to select other relevant articles-

·                    doi.org/10.1007/s10924-022-027425

·                    doi: 10.3390/biomimetics7040186

·                    doi.org/10.3390/polym13172905

·                    doi.org/10.3390/ma14144039

·                    10.21272/jnep.13(2).02033  

·                    There is no information about no. of samples subjected for testing (in line 96 three samples being said) its not sufficient. For any authentication its mandatory to have 6 samples studied.

             The work seems to be limited to only AFM analysis there are no analytical or simulation studies to back up the work for validation

             The authors have to exhaustively bring in the literature carried out on the dentistry by using AFM approach and at least add more dimensions in terms of comparison.

            Error plot has to be drawn in figure 8 and 9

            Results and discussions are acceptable only with any correlation between the current used material and existing materials in public domain

            There is no comparative study with analytical/simulation for validation of the extracted results

            Statistical analysis has to be explained with what percentage of confidence level is it 90% or 95%

            Deviation in the results need to be mentioned with standard deviation with ±tolerance indication with unit (many location the unit is not mentioned)

             Reference 17 has been used in most of the location of discussion, is it the only relevant article for comparison, it’s better to have tabular column with results discussed at a common variable such as AFM

             As per journal guidelines, the findings and their implications should be discussed in the broadest context possible and limitations of the work highlighted.

             As per journal guidelines, future research directions may also be mentioned. This section may be combined with Results

             The discussion section lags in explanation with respect to the work carried out.  there are no citations in discussion section to compare the work with existing materials

             Conclusion looks to be generic need to compile the outcomes and state based on the tests conducted and convey how best this can fit in the current context for any application.

             As per journal guidelines, this section is not mandatory but can be added to the manuscript if the discussion is unusually long or complex

             In conclusion section, values have to be displayed with explanation. It's better to mention the salient features of the entire work in terms of bullet points with current context

Round 2

Reviewer 1 Report

Dear Authors,

You have sufficiently revised the manuscript, thank you for your scientific contribution to this important topic. I believe in that point the article could be accepted.

Best regards

Reviewer 2 Report

The authors have addressed all the queries raised by reviewer. The manuscript may be accepted in its present form.